# The Role of Different Behavioral and Psychosocial Factors in the Context of Pharmaceutical Cognitive Enhancers’ Misuse

**DOI:** 10.3390/healthcare10060972

**Published:** 2022-05-24

**Authors:** Tina Tomažič, Anita Kovačič Čelofiga

**Affiliations:** Faculty of Electrical Engineering and Computer Science, Institute for Media Communications, University of Maribor, 2000 Maribor, Slovenia; anita_kovacic@hotmail.com

**Keywords:** pharmaceutical cognitive enhancement, nootropics, substance abuse, smart drugs

## Abstract

In an effort for better memory, greater motivation, and concentration, otherwise healthy individuals use pharmaceutical cognitive enhancers (PCEs), medicines for the treatment of cognitive deficits of patients with various disorders and health problems, to achieve greater productivity, efficiency, and performance. We examined the use of PCEs among 289 students at the Slovenian Faculty of Electrical Engineering and Computer Science in the behavioral and psychosocial context (students’ attitudes towards study, parents, health, leisure time, and work). Furthermore, we also addressed the immediate reasons, or the hypothesized connections of behavioral and psychosocial aspects, related to PCE misuse. The study consisted of a structured questionnaire, and chi-squared tests were used. An analysis of student statements revealed differences in students’ and parents’ attitudes toward good academic grades. In addition, students chose among 17 values related to relationships with parents, friends, partners, careers, study obligations, leisure, hobbies, material goods, appearance, and the future, and assessed their importance. Regardless of the group they belonged to, young people cited the same values among the most important. Good grades and parental opinions have proven to be key factors in the context of PCE abuse. This research was the first study to examine the relation between PCE misuse and the role of different behavioral and psychosocial factors.

## 1. Introduction

Pharmaceutical cognitive enhancers (PCEs), or nootropics, an umbrella term that refers to cognitive-enhancing smart drugs, stimulants, supplements, and other substances [1], are regulated prescription medicines that can improve or reduce the risk of disease in people with certain impairments or diseases. They were developed to treat cognitive dysfunction [2] and are prescribed for the treatment of several disorders, including narcolepsy, sleep disorder, attention deficit hyperactivity disorder (ADHD), schizophrenia [3], Alzheimer’s disease [4], dementia [5], etc. Most often, methylphenidate (Ritalin, Concerta), mixed amphetamine salts (Adderall), modafinil [6] and armodafinil (Provigil, Modiodal), piracetam (Oikamid, Kalicor), racetams, diazepam, alprazolam, etizolam, lorazepam, ephedrine, sertraline, fluoxetine, atomoxetine, morphine, oxycodone, bupropion, pyritinol (Encephabol), dihydroergocornine (Hydergine, Redergin), cinnarizine (Stugeron), xantinol-nicotinate (Complamin), and Meclofenoxate (Lucidril) are mentioned [7]. Benzodiazepines, opioids, selective serotonin reuptake inhibitors (SSRIs), and ephedrine are merged into PCEs, used by healthy people. Preclinical evidence for SSRIs as cognition enhancers justifies the use of mood, behavior, and cognition as outcome variables [8]. Separable modulations of responsive inhibition networks and the potential superiority of opioids are suggested to improve cognitive performance in healthy individuals [9]. It has also been proven that psilocybin therapy (ephedrine) increases cognitive and neural flexibility in patients with major depressive disorders and has a greater influence on healthy individuals [10]. Benzodiazepines have also been shown to increase alertness in healthy individuals [11]. According to Arce and Ehlers (2017), when used by healthy people with no specific disorders or diseases, nootropics, or PCEs, can strengthen or improve the state of cognitive abilities, such as enhancing learning, memory, attention, and concentration and raising alertness. According to Stein (2012), so-called “smart pills” will be used to promote learning and clarify thinking, “happy pills” to increase mood and improve temperament, and “pep pills” to increase energy and maximize motivation.

PCE prevalence estimates in different EU Member States range from 0.8% to 16%, depending on different factors, such as, for example, country, field of work, university, type, and purpose of the used drug [12,13,14,15,16,17]. In a 2008 study by Nature Magazine, 20% of 1400 participants responded that they already used a PCE as a cognitive enhancer [18]. Among German business newspaper users, the lifetime use of any drug for neuroenhancement was 88% [19]. Moreover, 5% of employees in a German insurance company used PCEs to enhance productivity or their mood [20]. In a survey of 1145 surgeons attending an International Conference, 8.9% of them used PCEs or drugs for cognitive enhancement [13].

The use of pharmacological cognitive enhancers has increased in recent years, even though such substances can have health-threatening side effects [21]. Many studies have been conducted recently, including in connection with psychiatric treatments. Some authors [21] have pointed out that attitudes towards cognitive enhancers are, in part, driven by dark personality traits (the Dark Triad of personality are Machiavellianism, narcissism, and psychopathy). On the other hand, they found out that a competitive climate may increase positive attitudes towards PCEs, but only in individuals scoring high on dark personality traits. A further study [22] has shown that the experience of anxiety, depression, perceived stress, and insomnia was related to increased reports of substance use. Poor self-confidence and self-medication were key motivators of illicit drug use in those presenting greater psychiatric distress. In another study [23], as the reported frequency of modafinil use increased, the number of perceived benefits increased, whilst the number of negative effects remained stable and unchanged. The respondents reported significantly more benefits than risks and more immediate benefits than longer-lasting benefits. Conversely, those with a reported psychiatric diagnosis perceived greater longer-lasting benefits compared with those without a psychiatric diagnosis.

Abuse of PCEs is more common in occupations with emphasized cognitive abilities and in workplaces where employees are required to pay more attention, focus, and be alert [24]. These are occupations where night, shift, or extended work is present and where work associated is with intense intellectual work, high responsibility, and stress. Thus, PCE use is more likely to occur among surgeons, nurses, pilots, air traffic controllers, firefighters, soldiers, drivers, and other occupational groups [25]. Comprehensive studies on the actual prevalence between occupational groups have not yet been carried out [24]. Due to the lack of empirical research, the motivations for the use of PCEs among employees can be speculated based only on broader social and work trends and research among student populations, which is [26] a key risk group for the non-prescribed use of these stimulant nootropic smart drugs, to improve their performance in their academic studies. On the one hand, it is undoubtedly encouraged by competition and the desire for the success of the individual, and, on the other hand, it depends on socio-economic factors and conditions and requirements in individual professional groups [27]. Finally, use is also associated with the general working population, which is supposed to use them to alleviate the effects of sleep deprivation and help cope with rising workloads [28].

The problem is more pronounced, especially in the student environment, as enhanced learning, memory, motivation, and concentration are important advantages in a highly competitive student environment. The use of PCEs among students does not usually serve as an aid to those who are already achieving a higher study average and would like to improve it further [29,30]. It is more common among those who abuse alcohol and illicit drugs, especially marijuana [13,14,15,28,31,32]. The proportion of users is higher among senior students [16,33] and more frequent in educational institutions with more demanding admission criteria and study requirements [29,34]. Students use PCEs because they believe they are an effective tool for studying; for non-academic purposes [35,36], they can improve focus/concentration and performance, improve memory and wakefulness, improve sleep, reduce anxiety and fear, and improve thinking; and for recreational reasons, they can support experimentation and increase creativity [14,28,37,38,39]. A further study has been conducted (Jensen), where students, PCE users, reported higher levels of stress and lower levels of ability to cope than the sample average. They preferred to use avoidant emotion-focused coping strategies until they were close to deadlines, where they then used stimulants as a problem-focused alternative coping strategy to moderate their stress. An important role in student use of PCEs is also played by parents, who, in many cases, are supposed to be aware that they enjoy them and even support or encourage them [32,40]. As Mehlman [40] writes, to improve cognitive abilities, parents should give PCEs to their children intentionally. Diller [41] wrote about Ritalin, which was given to children by ambitious parents, even on the recommendations of teachers who wanted more order in classrooms. Similarly, Farah et al. [42] add that the use of Ritalin for this purpose is already evident in the number of users among children, which, in some school districts, exceeds the average number of patients with ADHD for whom the drug is prescribed.

These are indirect reasons for PCE misuse, already covered by existing studies. In our research, we address the immediate reasons, or the hypothesized connections of behavioral and psychosocial aspects, related to PCE misuse, which has not been researched in any study so far.

In our research, we developed the following research questions:RQ1: What are the most important psychosocial factors among student PCE users?RQ2: What is the attitude among PCE users towards study, and what is the role of their parents?RQ3: Does the attitude towards psychosocial values differ between students who are users of PCEs compared to other students?

## 2. Methods

### 2.1. Participants and Procedures

The sample consisted of 289 students (39% female, 122; 61% male; 167) of the Faculty of Electrical Engineering and Computer Science in Maribor, Slovenia, with an age range from 18 to 34 years (M = 18, SD = 2.82). Due to the importance of students’ relationships with their parents, we obtained some data about parents as well. Among these participants’ parents, 58% (168) of the fathers and 45% (130) of the mothers had obtained a high school level of education or below, and 38% (110) of the fathers and 41% (119) of the mothers had received a graduate degree or above. The others did not want to answer about the level of education of their parents.

Students were informed that their responses to questionnaires would be kept anonymous and confidential, and the collected data would be used for academic research only. It took approximately 15 min to complete the questionnaires. The students participated in the survey on a voluntary basis and were not compensated for their participation.

### 2.2. Measures

The survey was conducted in paper-and-pencil form during random class hours, in agreement with the lecturer, who distributed and collected the surveys in the last half of 2019. The surveys were filled out by 289 students of the Faculty of Electrical Engineering and Computer Science in Maribor, Slovenia. Convenience sampling was used, which limited the conclusions of the study to a small group of students with specific characteristics. The exclusion criteria were self-reported medical conditions of students with a prescription for PCEs, i.e., students who received PCEs due to their medical condition. In the existing literature, we looked for factors that may affect the study area. This was also the starting point for designing the questionnaire. We adjusted the factors to the studied population and their way of life, especially the fact that they are students. The survey questionnaire consisted of questions with closed answers, first regarding the level of agreement with statements about different psychosocial factors. The survey collected demographic data, including date of birth, gender, and the highest level of parental education. Before using the questionnaire, it was reviewed by a methodologist and an expert in our field and then confirmed by the Ethics Commission. Before use, we tested it on a small number of students to see if it was demarcated (and contributed to the validity of the measurement). Content validity was ensured by the fact that the design of the questionnaire was based on the existing literature. Prior to the analysis, we also checked the normality of the data distribution. This unvalidated questionnaire can be a useful tool to provide the expected results because it makes sense to test different social groups to determine if PCEs are predominant among them. Additionally, because the factors that influence PCE abuse are huge, we tried to identify them in this way. Where data were not distributed normally, the Kruskal–Wallis test was used for comparison. If a difference was found in any of the items, the Mann–Whitney test (with Bonferroni correction) was performed for each group combination. Cronbach’s alpha was calculated on the items, measuring Q5, Q6, Q7, and Q9, to provide information on the internal consistency of the data. It indicates (α = 0.622) the almost acceptable level of reliability of the analyzed data. The main purpose of the study was to explore the psychosocial and behavioral factors among student PCE users. We defined “smart drugs” for participants and asked about the potential use of PCEs and the details of their experiences, which were filled out only by those who had already used PCEs. Subjects had to mark substances they had used for cognitive enhancement with substrate and trademark names. The questionnaire also contained space to add further substances.

Next, we addressed the behavioral and psychosocial context with questions about students’ attitudes towards study, parents, health, leisure time and work; for instance, their competitiveness, desire for success, demands of parents, learning habits, attitude to academic grades, sleeping and eating habits, and leisure activities.

The questionnaire included a set of 32 statements, applying to:-The attitude towards grades: how important are grades to students’ parents, how much effort students put into them, how much time they devote to learning, how and when they learn, what their personal beliefs are about the importance of good grades, and the impact of grades on the amount of pocket money;-The importance of students’ grades in comparison to their friends, careers, leisure activities and other obligations, and sports;-How healthy they are: adequate and quality sleep, nutrition, exercise, relaxation, concentration, and memory;-Their attitude towards studies: whether they find it challenging, boring, stressful, diverse, or energy intensive.

Furthermore, students assessed 17 values related to relationships with parents, friends, partners, careers, study obligations, leisure, hobbies, material goods, appearance, and their future, and assessed their importance.

## 3. Results

A total of 289 students signed the informed consent to participate and completed the survey. A total of 273 surveys were completed correctly. Three persons were excluded from participating in the study due to their medical condition, as they had been prescribed PCEs by psychiatrists. A total of 270 students met all the criteria for inclusion in the study. The mean age of these was 21 (±2.28) years.

The data were analyzed statistically. Descriptive statistics were calculated in order to provide basic statistical information about the distribution of the data. The normality of distribution of the data was tested with the Kolmogorov–Smirnov test. As the distribution of the data deviated statistically significantly from normal, a non-parametric test, the Kruskal–Wallis test, was used for the group differences analysis. The Mann–Whitney test was used as a post-hoc test.

Of the respondents, 4.8% (13) confirmed that they had already used a prescription medicine that had not been prescribed for any disorder to improve their cognitive abilities, mostly to improve focus, but the desire for creativity and alertness follows. The most misused substances were modafinil and armodafinil, which are among the most used (83%, 11). A further 31% (4) purchased PCEs from another person or from a foreign website (23%; 3).

The information that 46% (6) did not plan to misuse PCEs anymore is reassuring, but 38% (5) had not yet decided about this. There was also a low proportion of those who would recommend them to others, only 15% (2). A further 69% (9) were aware that PCEs are most likely to harm them, and 58% (7) said they should not be freely accessible.

Respondents were sorted into four groups: those who had never heard about PCEs (Group A), heard about them (B), knew someone who had tried PCEs (C), or those who had tried PCEs themselves (D). Such distribution of participants gave us a broader insight into students’ beliefs connected with PCE misuse. We decided to separate students who knew PCE users from those with friends using PCEs for non-medical purposes, considering the findings of Ford and Ong (2014) that college students with friends that use prescription stimulants non-medically are more likely to use PCEs as well, since they believe their use is acceptable or necessary to fit in with the social group. So, in the research, we connected different groups of students with their attitude to studies (grades, desire to prove themselves, study obligations, according to the characteristics of schooling) and parents, in connection with health care and leisure activities and work. An analysis of 32 statements revealed differences in students’ and their parents’ attitudes toward good grades and learning habits. There were no statistically significant differences in the analysis of other statements.

Grades had been identified to be associated positively with PCE misuse. The comparison of the groups showed differences regarding the importance of good grades for their parents and the importance of good grades to become what students want (Table 1).

When deciding on PCE misuse, parents’ attitude to grades also seemed to be important. Students who had heard of PCEs consider grades to be very important for their parents (compared to those who did not know about PCEs). Compared to those who had already tried PCEs, students who had heard of PCEs were more in favor of the claim that only with good grades can they become what they want (Table 2).

Differences between groups were also reflected in the students position on learning habits, such as learning during nighttime. Differences occurred between groups who were unfamiliar with PCEs (who rarely learned at night), compared to students who had already tried PCEs (Table 3).

Furthermore, students chose among 17 values related to relationships with parents, friends, partners, careers, study obligations, leisure, hobbies, material goods, appearance, and future, and assessed their importance. Regardless of the group they belonged to, young people cited the same values among the most important. Those who ranked love among the three most important factors placed it in the first place, regardless of the group to which they belonged. Those who had already tried PCEs ranked love in a smaller proportion in the first place.

Students ranked their parents’ praise first or second if this value was important to them. None of them were from the group that had already tried PCEs. Parental praise was most important to those who had not heard of PCEs, but this difference did not prove statistically significant for further analysis.

Academic grades for students who had already heard of PCEs, who knew someone who was taking them, or had already tried them on their own, appeared mostly in the first place, while those who had not heard of PCEs, in most cases, ranked them in second place in importance. Although the observed differences proved to be statistically insignificant, they complement our findings significantly.

Students who valued consideration of their parents mostly put this value in second place if they had not heard of or known about PCEs, in third place if they knew someone who had already tried PCEs, and in first place if they had tried them themselves. The observed differences between the groups did not prove to be statistically significant after further tests.

Students who care about not being bored ranked this value second in most groups (except for the group who knew someone taking PCEs; in this group, the value was distributed evenly in first and second place). Further analyses showed that the differences between the groups were not statistically significant.

Among students who valued hobbies and hobby activities highly, all groups ranked the latter in second place in most cases. Young people who ranked money among the three most important values ranked the latter in most cases, regardless of the group to which they belonged. Few respondents ranked other material goods among the most important values; these, however, mostly covered second place. Sport, as an important factor, was ranked second in most cases.

No participant from the group who knew anyone taking PCEs ranked appearance among the most important values. The number of young people for whom appearance was important was small; those unfamiliar with PCEs mostly ranked them second, those who had heard of PCEs ranked them first, and those who had already tried PCEs ranked them third. Differences between groups were not statistically significant.

Students who identified family relationships as an important factor in their lives put them in second place in most cases, except in the group that had already tried PCEs, and the problems took first place in the ranking. The chi-squared test showed that there were no statistically significant differences between the groups.

The feeling of freedom and exits, in most cases, found themselves in third place in terms of the importance of factors. They were ranked higher by young people who did not know about or had only heard of PCEs. The chi-squared test showed that there were no statistically significant differences between the groups.

The possibility of employment was mostly ranked 3rd as an important factor in life (it appeared in first place only in the group that did not know about PCEs). The chi-squared test showed that there were no statistically significant differences between the groups.

In many cases, the future was ranked third among young people who ranked it among the most important factors. It was ranked first only in the group that had already heard of PCEs. The chi-squared test showed that there were no statistically significant differences between the groups.

We were also interested in the attitude towards illicit drugs. Those who had already tried PCEs used marijuana often and cocaine occasionally, while other drugs were present to a lesser extent (see also Vicario et al., 2020). The same students also consumed alcohol and smoked cigarettes more often.

In our research we followed the following research questions:RQ1: What are the most important psychosocial factors among students PCE users? An analysis of 32 statements revealed differences in students’ and parents’ attitudes toward good grades and learning habits. There were no statistically significant differences in the analysis of other statements.RQ2: What is the attitude among PCE users towards study, and what is the role of their parents? Academic grades had been identified to be associated positively with PCE misuse. The comparison of the groups showed differences regarding the importance of good grades for their parents and the importance of good grades to become what students wanted. Those who had heard of PCEs felt that grades were more important to their parents. When deciding on PCE misuse, parents’ attitudes to grades also seemed to be important. Students who had heard of PCEs considered grades to be very important for their parents (compared to those who did not know about PCEs). Compared to those who had already tried PCEs, students who had heard of PCEs were more in favor of the claim that only with good grades can they become what they want. Differences between groups were also reflected in the students’ position on learning habits, such as learning during nighttime. Differences occurred between groups who were unfamiliar with PCEs (who rarely learned at night), compared to students who had already tried PCEs.RQ3: Does the attitude towards psychosocial values differ between students who are users of PCEs compared to other students? Students chose among 17 values related to relationships with parents, friends, partners, careers, study obligations, leisure, hobbies, material goods, appearance, and future, and assessed their importance. Regardless of the group they belonged to, young people cited the same values among the most important.

## 4. Discussion

Good academic grades and parents’ opinions about them have proven to be among key factors in the context of PCE misuse. Findings were consistent with the conclusion of Li [43], who pointed out academic pressure as one of the main sources of personal life pressure for adolescents. It seems PCEs are not an academic shortcut for those who want to achieve better results, which our findings suggest. PCE users have a lower grade point average [30,44]. Users of PCEs have a history of heavy alcohol use and illicit drug involvement [14,16,29,31,32,45], as also proved by our study. Perhaps also because of heavy drinking and drug misuse, students who engage in NPS for study purposes appear to be struggling academically [32,45]. PCE misuse is also related to a higher level of stress or perceived pressure to perform [14,29]. PCE misuse can be predicted in schools with overwhelming demands, in institutions with more competitive admission criteria [34], and for students nearing the end of an undergraduate degree course or at postgraduate level [16,31].

The non-medical use of PCEs certainly deserves in-depth consideration and weighty consideration of the benefits and consequences for both the individual and society. We believe that we can still do the most for our cognitive abilities with traditional or alternative methods, such as a healthy diet, sufficient and quality sleep, exercise in nature and physical exercise, relaxation, and meditation, with various strategies and techniques of mental training and, of course, with quality interpersonal relationships. According to Szot et al. [46], it is also important to have an adequate intake of minerals and vitamins, which also have a positive influence on cognitive functions. Furthermore, some authors [47] also advise the implementation of interventions using a multi-component whole-school approach and life skill curriculum as a preventive measure, to improve mental health outcomes and, consequently, reduce students’ chances of consuming PCEs later.

The results of this study should be interpreted in light of some limitations. The study results suggest that different behavioral and psychosocial aspects need to be considered within the PCE misuse research. Although a variety of behavioral and psychosocial predictors were included, other predictors could be tested as well, such as empathy, moral reasoning, creativity, and motivation [48]. The sample in this study was limited to a particular faculty. The number of included students was too small to generalize their views on PCE misuse in general. Furthermore, we found it difficult to separate the behavioral and psychosocial aspects of PCE misuse strictly, so we did not consider them separately.

We can indicate some theoretical and practical implications from our study. The use of PCEs among students does not usually serve as a benefit to those who are already achieving a higher study average and would like to improve it further. They use them because they believe they are a great tool for studying (such as improving memory, thinking, creativity, and concentration) and for non-academic purposes (such as improving sleep, reducing fear and anxiety). They are used more commonly by those students who abuse alcohol and illicit drugs. An important role in the use of PCEs by students is played by their parents, who, in many cases, are aware that they enjoy PCEs and even support and encourage them.

This research was the first study to examine the relation between PCE misuse and the role of different behavioral and psychosocial factors. The PCE misuse context remains relatively unexplored and refers mainly to connections with academic success, history of alcohol use and illicit drugs, demographic characteristics, parental attitudes toward PCEs, and competitive admission criteria in friendship. It is important to recommend that further research is needed, both in terms of the actual neurophysiological effects of PCEs and the prevalence and socio-cultural specifics of their use by different populations of individual national environments. According to Bortolato [6], we suggest an additional study to investigate the effects of modafinil in remediating cognitive dysfunction in major depressive disorder.

## Figures and Tables

**Table 1 healthcare-10-00972-t001:** Students’ attitude to academic grades.

Groups			Grades Important to Parents	Good Grades Are a Condition for Success
A	*n*	Valid	109	110
		Missing	3	2
	Mean		3.17	2.34
	Std. Deviation		1.113	1.229
	Min		1	1
	Max		5	5
B	*n*	Valid	80	80
		Missing	3	3
	Mean		3.69	2.56
	Std. Deviation		1.001	1.135
	Min		1	1
	Max		5	5
C	*n*	Valid	36	37
		Missing	1	0
	Mean		3.31	2.27
	Std. Deviation		1.215	1.239
	Min		1	1
	Max		5	5
D	*n*	Valid	9	13
		Missing	4	0
	Mean		2.78	1.62
	Std. Deviation		1.302	0.650
	Min		1	1
	Max		5	3

Those who had heard of PCEs felt that grades are more important to their parents.

**Table 2 healthcare-10-00972-t002:** Association of good academic grades and PCE misuse.

	Groups	*n*	Mean Rank	Sum of Ranks
Good grades are a condition for success	B	80	50.16	4013.00
	D	13	27.54	358.00
	Total	93		

**Table 3 healthcare-10-00972-t003:** Students’ position on learning habits.

Groups		*n*	Min	Max	Mean	Std. Deviation
A	I only study at night	111	1	5	2.72	1.363
	Valid *n* (listwise)	111				
B	I only study at night	81	1	5	3.05	1.303
	Valid *n* (listwise)	81				
C	I only study at night	37	1	5	3.16	1.344
	Valid *n* (listwise)	37				
D	I only study at night	12	1	5	3.83	1.337
	Valid *n* (listwise)	12				

## Data Availability

The datasets used and/or analyzed during the current study are available from the corresponding author on reasonable request.

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
