# Peer review of "The Role of Different Behavioral and Psychosocial Factors in the Context of Pharmaceutical Cognitive Enhancers’ Misuse"

_healthcare, 2022, doi:10.3390/healthcare10060972_

Round 1
Reviewer 1 Report
I am grateful for the work done by the authors, however, I must make a number of major points:
Regarding the introduction, this should be expanded and improved. I suggest that they take into consideration previous authors and studies that support their research proposal. In addition, this presentation of previous studies should be comprehensive and rigorous.
As for the methodological section, I would need to see more precision in the description of the measurement tool and the procedure for its application: is it a validated questionnaire, its validity and reliability index, the internal reliability of the sample, normal distribution, etc.? They should report the type of questions and answers by way of example.
In addition, the application process: How was it applied, when, in what form? The methodology section lacks relevant information. It is also useful to describe the type of sample taken for the study: convenience, incidental sampling, random, random with control...?
As for the results, these are descriptive and therefore exploratory in nature. After the results and discussion section (to be improved with clear comparison and discussion with previous relevant studies), the authors should clearly discuss the limitations of their study and, in addition, include a conclusion section showing the potential of their findings and their theoretical and practical implications.
I recommend addressing all these issues in order to proceed with the review of this research study.
Best regards.
Author Response
Responses to Reviewer 1:
- Regarding the introduction, this should be expanded and improved. I suggest that they take into consideration previous authors and studies that support their research proposal. In addition, this presentation of previous studies should be comprehensive and rigorous.
RESPONSE: Thank you for this comment. We have expanded and improved the Introduction, which comes as a great contribution to our paper.
- As for the methodological section, I would need to see more precision in the description of the measurement tool and the procedure for its application: is it a validated questionnaire, its validity and reliability index, the internal reliability of the sample, normal distribution, etc.? They should report the type of questions and answers by way of example. In addition, the application process: How was it applied, when, in what form? The methodology section lacks relevant information. It is also useful to describe the type of sample taken for the study: convenience, incidental sampling, random, random with control...?
RESPONSE: We thank the reviewer for noticing that our methodology is poor. We have added the description of the measurement tool and the procedure and the type of sample, which can contribute greatly to the quality of the manuscript.
- As for the results, these are descriptive and therefore exploratory in nature. After the results and discussion section (to be improved with clear comparison and discussion with previous relevant studies), the authors should clearly discuss the limitations of their study and, in addition, include a conclusion section showing the potential of their findings and their theoretical and practical implications.
RESPONSE: We thank the reviewer for the idea to improve the Conclusion section. We have added some missing points in the conclusion, as practical implications, and reorganized the limitations, and deleted some necessary text.
Reviewer 2 Report
Major revision.
- The title does not reflect the core of the study. Please be more specific, as the study touches on specific areas studied in a narrow and specific group.
- Explain the abbreviation "nxy" - in the line 31
- Please post the referring citation to the stated list of medicines - line 39
- Please provide the background for benzodiazepines, opioids, SSRIs, and ephedrine to become named PCEs in this article.
- Please post the referring citation to the stated opinions - line 56
- Please provide more specific information on the inclusion and exclusion criteria - line 117.
- What was the response rate?
- The questionnaire should be attached to the supplementary files with the referral within the manuscript section Methods.
- Explain how this unvalidated questionnaire may be a useful tool to provide the expected results and become a background to provide the conclusions.
- Describe the details of statistical methods and the reason why where they have been chosen for analysis of these data.
- Try to make Table 1 more readable. Consider using abbreviations and legends below the table - line 179.
- "When deciding for PCE misuse, parents' attitude to grades also seemed to be important. " - what was p-value for this result? - line 181.
- Adjust all the results to the same question as at my point nr 11.
- line 271 - there is a citation numbered 34 referring to the first sentence of the discussion summarizing the study. In my opinion, this citation is not needed here.
- The study cannot suggest anything, the results may - line 297.
- "Regarding a better understanding of the 318 trend, according to Tomažič and Kovačič [18]" refers to the self-cited paper of both authors of the reviewed paper - and cannot be enough to support the same author's statement - line 319
- Citation nr 2 has the wrong DOI number
Author Response
Responses to Reviewer 2:
- The title does not reflect the core of the study. Please be more specific, as the study touches on specific areas studied in a narrow and specific group.
RESPONSE: Thank you for recognizing this disadvantage in the title. We have changed the title to be more specific.
- Explain the abbreviation "nxy" - in the line 31
RESPONSE: Thank you for recognizing this. It was a mistake and we have deleted nxy.
- Please post the referring citation to the stated list of medicines - line 39
RESPONSE: Thank you for recognizing that the citation was missing.
- Please provide the background for benzodiazepines, opioids, SSRIs, and ephedrine to become named PCEs in this article.
RESPONSE: Thank you for recognizing this gap in the study. We have provided the background for these substances in the Introduction section.
- Please post the referring citation to the stated opinions - line 56
RESPONSE: Thank you for recognizing that the citation was missing. We have added it.
- Please provide more specific information on the inclusion and exclusion criteria - line 117.
RESPONSE: Thank you, we have added the explanation.
- What was the response rate?
RESPONSE: Regarding the response rate, we explained it in the Results section. But, more specifically, 273 students completed the survey correctly. We considered 270 surveys, as three students took PCE for medical reasons, so they were excluded from participating in the study due to their medical condition, as they had been prescribed PCE by Psychiatrists.
- The questionnaire should be attached to the supplementary files with the referral within the manuscript section Methods.
RESPONSE: thank you for your suggestion. We attached the questionnaire.
- Explain how this unvalidated questionnaire may be a useful tool to provide the expected results and become a background to provide the conclusions.
RESPONSE:
Thank you for this valuable question. We believe that this unvalidated questionnaire can be a useful tool to provide the expected results because: The factors that influence PCE abuse can be immense, and we have tried to identify them in this direction. The questionnaire could become a background to provide conclusions.
- Describe the details of statistical methods and the reason why where they have been chosen for analysis of these data.
RESPONSE: Thank you for this comment. We have improved the Methodology, which comes as a great contribution to our paper. The data was analyzed statistically. Descriptive statistics were calculated in order to provide basic statistical information about the distribution of the data. The normality of distribution of the data was tested with Kolmogorov-Smirnov test. As the distribution of the data deviated statistically significantly from normal, a nonparametric test – the Kruskal-Wallis test, was used for the group differences analysis. The Mann-Whitney test was used as a post-hoc test.
- Try to make Table 1 more readable. Consider using abbreviations and legends below the table - line 179.
RESPONSE: Dear reviewer, thank you for recognizing that Table 1 should me more readable. We shortened the statements written in the Table and have done some other abbreviations, which contributed greatly to the readability of the Table.
- "When deciding for PCE misuse, parents' attitude to grades also seemed to be important. " - what was p-value for this result? - line 181.
RESPONSE: The p-value for this result is 0.006. We attach the Table: Results of the Kruskal-Wallis test in Question 5
- Adjust all the results to the same question as at my point nr 11.
RESPONSE: Thank you, we have shortened and used abbreviations and now all the Tables are much more readable.
- line 271 - there is a citation numbered 34 referring to the first sentence of the discussion summarizing the study. In my opinion, this citation is not needed here.
RESPONSE: thank you for recognizing this. We absolutely agree and we have deleted it.
- The study cannot suggest anything, the results may - line 297.
RESPONSE: Thank you so much! We absolutely agree and corrected the word usen.
- "Regarding a better understanding of the 318 trend, according to Tomažič and Kovačič [18]" refers to the self-cited paper of both authors of the reviewed paper - and cannot be enough to support the same author's statement - line 319
RESPONSE: Thank you so much, it’s absolutely true, so we deleted the citation.
- Citation nr 2 has the wrong DOI number
RESPONSE: Thank you for noticing our mistake. We have corrected it and added the right DOI number.
Round 2
Reviewer 1 Report
I thank the authors for their work, but there are still serious areas for improvement. The introduction is still very weak. A paragraph has been added but it still does not show adequate background and theoretical conceptualisation. This needs to be improved considerably.
In the methods section, aspects of reliability and robustness of the sample and the questionnaire used are still not shown. Indicating that the existing literature was followed is not sufficient to validate an instrument. Some alpha analysis of the questions and sample should be shown. Furthermore, I insist, how was the tool designed, were experts consulted, was an external critical review carried out? Everything related to the measuring instrument, its design, validity and application should be indicated in detail.
I suggest that these questions be addressed by the authors.
Best regards.
Author Response
1. The introduction is still very weak. A paragraph has been added but it still does not show adequate background and theoretical conceptualisation. This needs to be improved considerably.
RESPONSE:
Thank you so much for this idea to improve the Introduction. The Introduction was poor, so we added more background and further theoretical conceptualization.
2. In the Methods section, aspects of reliability and robustness of the sample and the questionnaire used are still not shown. Indicating that the existing literature was followed is not sufficient to validate an instrument. Some alpha analysis of the questions and sample should be shown. Furthermore, I insist, how was the tool designed, were experts consulted, was an external critical review carried out? Everything related to the measuring instrument, its design, validity and application should be indicated in detail.
RESPONSE:
Dear reviewer, thank you for this valuable comment. We have added and extended the Methods section.
Convenience sampling was used, which makes the conclusions of the study limited to the small group of students with specific characteristics.
The Cronbach's alpha was calculated on the items, measuring Q5, Q6, Q7 and Q9, to provide information on the internal consistency of the data. It indicates (α = 0,622) the almost acceptable level of reliability of the analyzed data.
In the existing literature, we looked for factors that may affect the study area. This was also the starting point for designing the questionnaire. We adjusted the factors to the studied population and their way of life, especially the fact that they are students.
Before using the questionnaire, it was reviewed by a methodologist and an expert in our field, and then confirmed by the Ethics Commission.
Reviewer 2 Report
The Authors have largely addressed my concerns and issued revisions.
Author Response
The Authors have largely addressed my concerns and issued revisions.
RESPONSE:
Thank you for all the effort and the time taken to evaluate our paper. We are glad that you recognized the better and improved version of the paper.